# Attention-Guided Transfer Learning for Identification of Filamentous Fungi Encountered in the Clinical Laboratory

Tsi-Shu Huang,[a] Kevin Wang,[b] Xiu-Yuan Ye,[a] Chii-Shiang Chen,[a] Fu-Chuen Chang[b]

aDivision of Microbiology, Department of Pathology and Laboratory Medicine, Kaohsiung Veterans General Hospital, Kaohsiung, Taiwan
bDepartment of Applied Mathematics, National Sun Yat-sen University, Kaohsiung, Taiwan

Tsi-Shu Huang and Kevin Wang are co-first authors. Author order was determined both alphabetically and in order of increasing seniority.

**ABSTRACT** This study addresses the challenge of accurately identifying filamentous fungi in medical laboratories using transfer learning with convolutional neural networks (CNNs). The study uses microscopic images from touch-tape slides with lactophenol cotton blue staining, the most common method in clinical settings, to classify fungal genera and identify *Aspergillus* species. The training and test data sets included 4,108 images with representative microscopic morphology for each genus, and a soft attention mechanism was incorporated to enhance classification accuracy. As a result, the study achieved an overall classification accuracy of 94.9% for four frequently encountered genera and 84.5% for *Aspergillus* species. One of the distinct features is the involvement of medical technologists in developing a model that seamlessly integrates into routine workflows. In addition, the study highlights the potential of merging advanced technology with medical laboratory practices to diagnose filamentous fungi accurately and efficiently.

**IMPORTANCE** This study utilizes transfer learning with CNNs to classify fungal genera and identify *Aspergillus* species using microscopic images from touch-tape preparation and lactophenol cotton blue staining. The training and test data sets included 4,108 images with representative microscopic morphology for each genus, and a soft attention mechanism was incorporated to enhance classification accuracy. As a result, the study achieved an overall classification accuracy of 94.9% for four frequently encountered genera and 84.5% for *Aspergillus* species. One of the distinct features is the involvement of medical technologists in developing a model that seamlessly integrates into routine workflows. In addition, the study highlights the potential of merging advanced technology with medical laboratory practices to diagnose filamentous fungi accurately and efficiently.

**KEYWORDS** convolutional neural network, transfer learning, attention, filamentous fungi

Address correspondence to Fu-Chuen Chang, changfc@math.nsysu.edu.tw.
The authors declare no conflict of interest.

Filamentous fungi spread ubiquitously in our environment; invasive infections caused by these microorganisms are rising in immunocompromised patients (1). The risks for the development of severe fungal infections include patients undergoing blood and marrow transplantation, solid-organ transplantation, and major surgery (especially gastrointestinal surgery); patients with AIDS, neoplastic disease, and advanced age; patients receiving immunosuppressive therapy; and premature infants (2–5). In addition, *Aspergillus* species are a significant cause of opportunistic mycoses and invasive mycosis (6, 7). Identifying *Aspergillus* species is critical because it provides vital information to guide the initiation of antifungal therapy (8–10) and offers clues about the disease spectrum.

Traditional laboratory diagnostic methods for invasive fungal infections include isolating the organism, serologic detection of antigens or antibodies, or histopathologic

evidence of invasion. However, isolation and identification are labor-intensive and require experienced staff. Therefore, in addition to traditional methods, PCR (11, 12), and matrix-assisted laser desorption ionization–time of flight mass spectrometry (MALTI-TOF MS) are reliable for detecting and identifying fungal pathogens but are not commonly used for mold identification in hospital-based labs. Despite its limitations, MALDI-TOF MS remains an indispensable tool in medical mycology. It is highly efficient for rapidly and routinely identifying mold isolates in clinical laboratories. However, the identification process involves complex extraction procedures and time-consuming steps, which may result in low diagnostic accuracy scores. Nonetheless, the clinical benefits of MALDI-TOF MS cannot be denied. Unfortunately, the current reference spectra library for medical mycology is inadequate, which limits its clinical utility (13, 14). Therefore, identifying filamentous fungi still relies heavily on the morphological characteristics of the microscopic images and the colony morphology. This method requires well-trained personnel, and subject to human errors, conclusions based on unreliable technical foundations can become arbitrary. As a result, identifying filamentous fungi in clinical laboratories remains challenging. They are generally recognized at the genus level, and only a few medically important ones are subsequently identified as species.

Deep learning, a subfield of machine learning that is a subpart of artificial intelligence, has been applied in the classification of medical imaging (15–18), including bacteria (19–21), yeast or yeast-like organisms (22), and parasites (18, 23, 24), so it stands to reason that deep learning might also assist in identifying filamentous fungi. Deep learning neural networks are made of multiple perceptrons with an input layer and many hidden layers before the output layer (25). A perceptron is a simple neural network that takes input as a data set and produces an output as either a classification, a category, or an outcome prediction. Convolutional neural networks (CNNs), a class of deep learning model, have been recognized as an effective technique for a wide range of computer vision tasks (16, 26) applied to facial recognition, object identification, image generation, and analysis. The advantage of these networks is that they take the entire image as the input and learn to extract essential features for classification during training.

CNN requires a large volume of labeled training data, the bottleneck of medical image classification. Transfer learning with attention-guided mechanisms can be utilized to overcome this hurdle. Transfer learning is learning a new task by transferring knowledge from a related task that has already been learned in big data sets, such as ImageNet, a large data set consisting of 14 million labeled images and more than 1,000 classes (27). In transfer learning, the CNN model is pretrained on ImageNet and then modifies the features (connection weights in the CNN) learned from the pretrained CNN model (i.e., fine-tuning) and applied to perform a new classification task (28). The advantage of transfer learning is improved accuracy and reduced training time since a new training model does not have to be built from scratch. In addition, previous studies have been successfully applied and showed that the features learned from natural images could be transferred to medical images, even if the target images significantly differ from the pretrained source images (29, 30). Applying attention mechanisms further improves the performance of transfer learning with CNNs. (31) The CNN model focuses on the whole input image. In contrast, the attention mechanism focuses on the target region in the input image, which is more significant to the classification results than the rest of the regions. The combined use of transfer learning and attention mechanism allows the model to solve the problem of data acquisition difficulty of fungal images and improve the weight of the critical fungal image features.

In some preliminary reports, the CNN transfer learning model has been successfully applied to diagnose filamentous fungi. Most of the previous reports focused only on the genus *Aspergillus* and used macroscopic patterns (32, 33). Microscopic morphologies' characteristics are critical for accurately identifying filamentous fungi and are a crucial determining factor for fungus identification since macroscopic patterns might

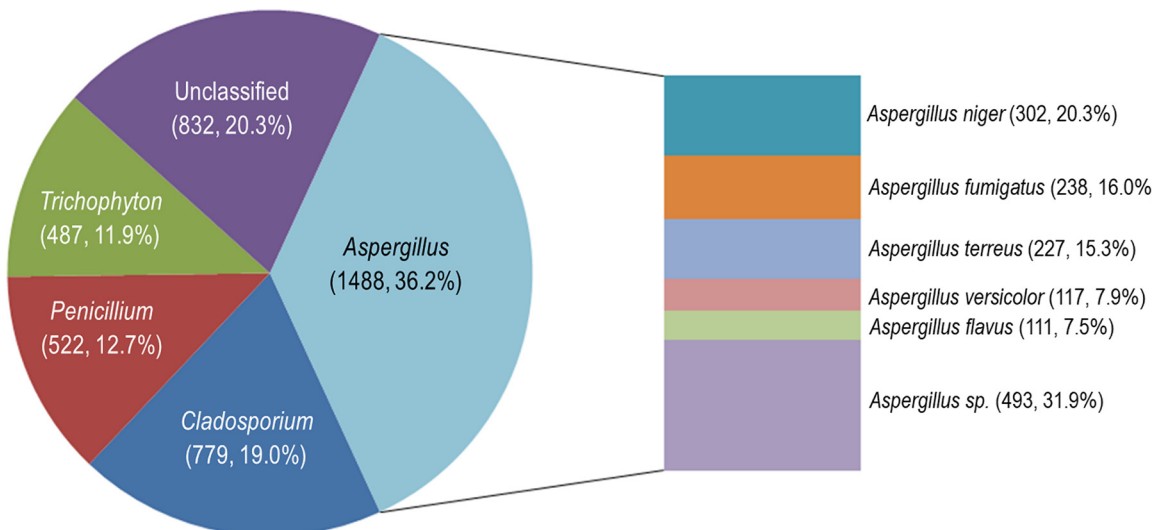

**FIG 1** The distribution of the 4,108 fungal images of common genera and six *Aspergillus* spp. in the data set.

be similar between different (species or genera) of fungi. Billones et al. (34) proposed a deep learning model to differentiate the fungal species with a data set consisting of 4,545 microscopic images containing 9 *Aspergillus* species and achieved 87.50% accuracy in training and 95.65% accuracy in validation. The images were obtained from the slide preparation stained with the glycerol-alcohol technique under a compound and stereo microscope, which is not the standard method used in a clinical laboratory. Ma. et al. developed an automated system based on deep learning to differentiate seven *Aspergillus* species (35). The fungal strains were prepared and inoculated with a specific number of conidia ($3 \times 10^4$). Images containing the typical representative morphology of conidiophores or colonies of each strain were from colonies on culture plate scanning using a dissecting microscopy (DM)/stereomicroscope platform. The detection accuracy was 98.2%. However, the technique is challenging to implement in the clinical laboratory.

Here we describe an application of transfer learning with CNNs for fungi classification as genera and identification as species for the genus *Aspergillus*. We used the microscopic images from touch-tape preparation and the lactophenol cotton blue (LPCB) staining technique available in every clinical laboratory. By utilizing CNNs with attention-guided transfer learning, our study accomplished an overall classification accuracy of 94.9% for four frequently encountered genera and 84.5% for *Aspergillus* species.

## RESULTS

The transfer learning models were built with Python 3.7.13 (36), TensorFlow 2.8.2, and Scikit-learn 1.0.2 in Google Colab and Kaggle.

**Data set.** A total of 4,108 images were used in this study. The pie chart in Fig. 1 displayed the distribution of the most common genera. *Aspergillus* was the most represented genus, with 1,488 images (20.3%), 837 of which were clinical specimens. The top five *Aspergillus* species were *A. niger* (302, 20.3%), *A. fumigatus* (238, 16.0%), *A. terreus* (227, 15.3%), *A. versicolor* (117, 7.9%), and *A. flavus* (111, 7.5%). The 74 other *Aspergillus* species accounted for 31.9% (*n* = 493). *Cladosporium*, *Penicillium*, and *Trichophyton* were also well represented, with 779 (19.0%), 522 (12.7%), and 487 (11.9%) images, respectively. Finally, there were 832 (20.3%) images from 56 other genera of filamentous fungi.

**Comparison of models with and without the attention mechanism.** We evaluated the performance of five CNN models for our fungal image classification task. The models achieved high training accuracies for classifying genera, ranging from 94.4% to 99.8% with the attention mechanism and 76.6% to 98.9% without it (as presented in Table 1). The addition of an attention mechanism improved training accuracy by −3.4%

**TABLE 1** Performance comparison of five transfer learning models with and without an attention mechanism for the classification of genera during model training and validation[a]

| Network configuration | Without attention | | With attention | |
|---|---|---|---|---|
| | Train accuracy (%) | Train loss | Train accuracy (%) | Train loss |
| ResNet50 | 94.3 | 0.170 | 96.4 | 0.109 |
| DenseNet201 | 97.8 | 0.078 | 94.4 | 0.183 |
| MobileNetV3 Large | 76.6 | 1.798 | **99.8** | 0.314 |
| EfficientNetV2M | 98.2 | 0.074 | 99.5 | 0.175 |
| Xception | **98.9** | **0.033** | 99.6 | **0.014** |

[a]Training accuracy, loss value, and computation time were measured during model training and validation to classify fungal genera. Bold text indicates the best value among the models.

to 23.2%. MobileNetV3 Large with an attention mechanism performed the best, while EfficientNetV2M and Xception models showed similar accuracies under both conditions. Additionally, Xception exhibited the lowest training loss under both conditions.

We used a similar method to train models for identifying different *Aspergillus* species within the same genus. Among the five models, Xception achieved the highest accuracy, up to 98.6% and 97.0%, respectively, with and without the attention mechanism. Moreover, the Xception model had the best loss values compared to the other models (see Tables 2 and 3).

**Performance metrics and evaluation of test data. (i) Classification of common genera.** When evaluating the effects of various CNN models on our fungal image classification task, we observed that the test accuracies for each genus ranged from 89.0% to 94.9% with attention and 73.9% to 92.8% without attention, demonstrating the model's strong ability to distinguish between the four common genera. Adding an attention mechanism resulted in a 21% improvement in the test accuracy of MobileNetV3 Large. However, DenseNet201, when incorporating attention mechanisms, actually decreased its performance by 3.6%. The accuracy decrease may be due to DenseNet201's already strong feature extraction and connectivity capabilities, leading to potential overfitting when attention mechanisms are introduced. MobileNetV3 Large achieved the best performance across all metrics when using attention mechanisms. (See Tables 4 and 5 for detailed results.)

Although the Xception model displayed lower train and test loss than the MobileNetV3 Large model, the latter proved to be superior across all performance metrics for mold genus classification, including F1 score, precision, and recall. This inconsistency suggests that the Xception model may be overfitting, which can be evaluated by examining its loss on training and test sets. While the Xception model's low loss may seem promising, its ability to accurately classify new and unseen data may be compromised due to overfitting on the training data. In contrast, the MobileNetV3 Large model appears to have better generalization

**TABLE 2** Summary of the network architecture and its parameters

| Layer (options) | Output shape | No. of parameters |
|---|---|---|
| Input-1 | None, 224, 224, 3 | 0 |
| Xception | None, 7, 7, 2,048 | 20 861 480 |
| BatchNormalization(name='BatchNormalization') | None, 7, 7, 2,048 | 8 192 |
| Conv2d_4(filters = 64, kernel_size = (1,1), padding = 'same', activation = 'relu') | None, 7, 7, 64 | 131 136 |
| Conv2d_5(filters = 32, kernel_size = (1,1), padding = 'same', activation = 'relu') | None, 7, 7, 32 | 2,080 |
| Conv2d_6(filters = 16, kernel_size = (1,1), padding = 'same', activation = 'relu') | None, 7, 7, 16 | 528 |
| Conv2d_7(filters = 1, kernel_size = (1,1), padding = 'valid', activation = 'sigmoid') | None, 7, 7, 1 | 17 |
| Conv2d_8(filters = 1, kernel_size = (1,1), padding = 'valid', activation = 'sigmoid') | None, 7, 7, 2,048 | 2,048 |
| Multiply(['conv2d_8','BatchNormalization'], name = 'Multiply') | None, 7, 7, 2,048 | 0 |
| Global_avg_pooling2d('Multiply') | None, 2,048 | 0 |
| Global_avg_pooling2d_1('conv2d_8') | None, 2,048 | 0 |
| RescaleGAP('global_avg_pooling2d'/ 'global_avg_pooling2d_1') | None, 2,048 | 0 |
| Dropout(rate = 0.5) | None, 2,048 | 0 |
| Dense(units = 128, activation = 'relu') | None, 128 | 262,272 |
| Dropout_1(rate = 0.25) | None, 128 | 0 |
| Dense_1(units = 5, activation = 'softmax') | None, 5 | 645 |

**TABLE 3** The performance comparison of five transfer learning models with and without an attention mechanism for the classification of *Aspergillus* species during model training and validation[a]

| | Without attention | | With attention | |
|---|---|---|---|---|
| Network configuration | Train accuracy (%) | Train loss | Train accuracy (%) | Train loss |
| ResNet50 | 67.8 | 0.878 | 95.5 | 0.131 |
| DenseNet201 | 78.8 | 0.654 | 97.4 | 0.080 |
| MobileNetV3 Large | 58.9 | 1.196 | 55.2 | 1.407 |
| EfficientNetV2M | 96.2 | 0.101 | 97.4 | 0.068 |
| Xception | **97.0** | 0.089 | **98.6** | **0.041** |

[a]Training accuracy, loss value, and computation time were measured during model training and validation to classify *Aspergillus* species. Bold text indicates the best value among the models.

performance. Moreover, incorporating an attention mechanism resulted in the highest accuracy of genus classification for the *Aspergillus* genus in most models, likely due to the distinctive club-shaped vesicles present at the apex of conidiophores in *Aspergillus* species, making them easy to differentiate from other genera. Although *Penicillium* also displays prominent morphological features under the microscope, this study observed some misclassification of images as *Aspergillus*, resulting in a lower accuracy rate. The confusion matrix summarizes the results by tabulating the correct and incorrect classifications. Table 6 compares the predicted and actual classes of the test set data. The sensitivity ranged from a high of 98.1% for the *Aspergillus* genus to a low of 90.6% for the "unclassified" group. Figures 2 and 3 display representative images from the training data set that were either correctly or incorrectly identified.

**(ii) Classification of *Aspergillus* species.** The MobileNet model performed the worst among the five test models for *Aspergillus* species classification with CNN, achieving an accuracy rate of approximately 53%, regardless of the use of the attention mechanism. This outcome highlights that CNN architectures can yield different results depending on the classification task and the techniques and strategies employed during network design and training. In general, models with attention performed better than those without, improving accuracy by 0.4% to 19.5%. Although the EfficientNetV2M model had similar performance without using an attention mechanism, the Xception model proved to be the superior choice, achieving the highest accuracy with attention on test data at 84.5% and a precision of 84.0%. (Tables 7 and 8).

The performance difference between the MobileNetV3 Large, EfficientNetV2M, and Xception models when classifying mold genera and *Aspergillus* species may be related to the structural characteristics of the molds. MobileNetV3 Large has a lightweight architecture, which may not be as proficient in capturing complex and diverse structural features necessary for distinguishing among *Aspergillus* species. However, it can effectively differentiate between different genera such as *Aspergillus*, *Penicillium*, *Cladosporium*, and *Trichosporon* since they exhibit significant differences in their morphological characteristics, which MobileNetV3 Large can detect better. On the other hand, EfficientNetV2M and Xception, which are deeper and more complex architectures, may be better suited to capture the complex structural features of *Aspergillus* species, leading to superior performance in *Aspergillus* species classification.

**TABLE 4** Performance metrics of five transfer learning modes compared for classifying five genera using a five-fold stratified cross-validation[a]

| | Without attention | | | | | With attention | | | | |
|---|---|---|---|---|---|---|---|---|---|---|
| Model | Test accu. (%) | Test loss | Macro F1-score (%) | Prec. (%) | Sens. (%) | Test accu. (%) | Test loss | Macro F1-score (%) | Prec. (%) | Sens. (%) |
| ResNet50 | 88.8 | 0.364 | 87.4 | 87.9 | 87.2 | 89.9 | 0.330 | 88.6 | 89.2 | 88.3 |
| DenseNet201 | 92.6 | 0.287 | 92.1 | 92.8 | 91.5 | 89.0 | 0.345 | 87.1 | 89.1 | 86.4 |
| MobileNetV3 Large | 73.9 | 1.657 | 68.2 | 78.6 | 67.5 | **94.9** | 0.218 | **94.4** | **94.8** | **94.0** |
| EfficientNetV2M | 92.6 | 0.254 | 91.8 | 92.4 | 91.5 | 93.6 | 0.225 | 93.6 | 93.5 | 91.9 |
| Xception | **92.8** | **0.245** | **92.1** | **92.6** | **91.8** | 94.4 | **0.195** | 93.9 | 94.2 | 93.6 |

[a]The best value among the models is highlighted in bold text. Prec., precision; Sens, sensitivity.

**TABLE 5** Accuracy of classifying five genera[a]

| Model | Accuracy without attention (%) | | | | | Accuracy with attention (%) | | | | |
|---|---|---|---|---|---|---|---|---|---|---|
| | A | C | P | T | U | A | C | P | T | U |
| ResNet50 | 93.5 | 88.6 | 82.0 | 84.0 | 87.8 | 94.2 | 90.9 | 80.8 | 86.7 | 88.8 |
| DenseNet201 | 94.9 | 91.2 | 90.3 | **88.5** | **92.9** | 94.6 | 92.7 | 80.4 | 75.2 | 89.1 |
| MobileNetV3 Large | 94.5 | 88.6 | 50.5 | 54.2 | 49.8 | **98.1** | 96.3 | **92.5** | **92.6** | 90.6 |
| EfficientNetV2M | **96.0** | 93.8 | **91.2** | 86.2 | 90.2 | 97.7 | 94.9 | 91.2 | 83.4 | 92.4 |
| Xception | 95.3 | **95.6** | 89.1 | **88.5** | 90.4 | 96.6 | **96.8** | 91.0 | 91.8 | **92.1** |

[a]The best value among the models is highlighted in bold text. A, *Aspergillus*; C, *Cladosporium*; P, *Penicillium*; T, *Trichophyton*; U, unclassified.

The test set data for *Aspergillus* species is displayed in Table 9, which presents the confusion matrix. Notably, *A. terreus* had the highest sensitivity rate at 93.0%, while *A. flavus* had the lowest rate at 73.0%. *A. terreus* and *A. niger* have distinct morphological structures compared to *A. flavus*, *A. fumigatus*, and *A. versicolor*. For example, *A. terreus* is known for having a unique structure with conidia produced in long chains, while *A. niger* has dark-colored conidiophores and distinctive flask-shaped fruiting bodies called ascomata. These easily recognizable features make it easier for CNNs to classify images of these two species accurately. In contrast, the other three species have less distinct morphological features, making it more challenging for CNNs to classify images of these species accurately.

## DISCUSSION

To our knowledge, no prior research has employed CNNs to classify filamentous fungal genera using microscopic morphology images from touch-tape technique slides, which remains the most commonly used and straightforward method in a clinical setting. Therefore, this study applied attention-guided transfer learning with CNNs for fungal image classification. The findings revealed that, with the integration of an attention mechanism, the MobileNetV3 Large model achieved the highest accuracy, resulting in an overall classification accuracy of 94.9% across the four commonly encountered genera. On the other hand, for *Aspergillus* species classification, the Xception model demonstrated the best performance, delivering an overall accuracy of 84.5%.

When utilizing deep learning models for classification tasks, it is crucial to consider multiple factors that can impact the model's performance, including data quality, model architecture, and hyperparameter tuning. Therefore, conducting a thorough analysis and making necessary adjustments are essential for achieving optimal results. Our study discovered that MobileNetV3, a lightweight CNN architecture, is suitable for classifying genera. However, for classifying *Aspergillus* species, which exhibit subtle differences between species, more complex models such as Xception, DenseNet201, and EfficientNet may be required to learn more complex features and deeper semantic representations.

Morphologic identification of fungi remains the foundation of any clinical mycology laboratory. Microscopic analysis for characteristic fungal structures is one of the critical parts of fungal identification. However, a limitation of morphology-based identification methods is that overlapping morphological characteristics inevitably lead to misidentification. Future research could incorporate additional features, such as colony morphology at various incubation times and characteristics of different culture media, including surface

**TABLE 6** Confusion matrix of genus classification results of the test set for MobileNetV3 Large model

| Genera | Predicted classification (n) | | | | | Sensitivity (%) | Specificity (%) |
|---|---|---|---|---|---|---|---|
| | *Aspergillus* (n = 1,532) | *Cladosporium* (n = 772) | *Penicillium* (n = 502) | *Trichophyton* (n = 485) | Unclassified (n = 817) | | |
| *Aspergillus* (n = 1,488) | 1459 | 2 | 4 | 5 | 18 | 98.1 | 97.2 |
| *Cladosporium* (n = 779) | 8 | 750 | 2 | 8 | 11 | 96.3 | 99.3 |
| *Penicillium* (n = 522) | 22 | 3 | 483 | 3 | 11 | 92.5 | 99.5 |
| *Trichophyton* (n = 487) | 4 | 3 | 6 | 451 | 23 | 92.6 | 99.1 |
| Unclassified (n = 832) | 39 | 14 | 7 | 18 | 754 | 90.6 | 98.1 |

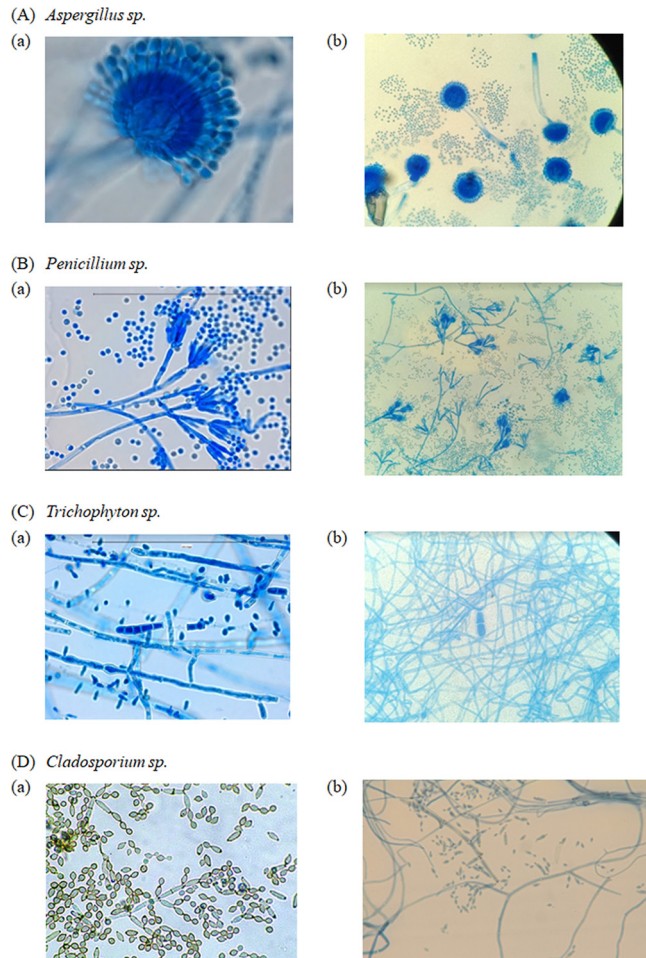

**FIG 2** (A to D) Representative figures of *Aspergillus* (A), *Cladosporium* (B), *Penicillium* (C), and *Trichophyton* (D) in the data set. The images are obtained from two sources: (a) Google and (b) photomicrographs of touch-tape preparation of fungal culture stained with LPCB and taken using a high-power objective (40×) on a compound light microscope.

and reverse colony color, to address this challenge. These added features could potentially improve classification accuracy in morphology-based identification methods.

AI analysis involves examining a vast array of raw data sets using predetermined parameters set by scientists. In this study, several CNN architectures were trained to perform classification tasks to determine the optimal configuration that matches the results of the experts. Additionally, since AI only recognizes digital values, human bias does not influence it, which can impact image interpretation. Another notable strength of this study is including an "unclassified" group, a crucial feature in real-world scenarios. In the future, if CNN image classification technology is coupled with automated scanning and image-capturing equipment for slides, multiple images of the same slide may be captured. However, many of these images may not have the distinctive characteristics necessary for identifying a specific mold and will likely be classified under the "unclassified" group. Ultimately, the conclusive identification of the slide is based on a comprehensive evaluation of all images acquired from the same slide.

A limitation of this study is the absence of clinical validation. Therefore, it is necessary to have an external validation data set that can be compared to a predicted reference standard based on the method used to evaluate how the model would perform in a clinical setting. It is also essential to perform clinical validation to detect failures that may not have been apparent during the model's development. Therefore, validating the model clinically before implementing it in clinical practice is imperative. We plan to collect additional samples to expand our clinical validation data set.

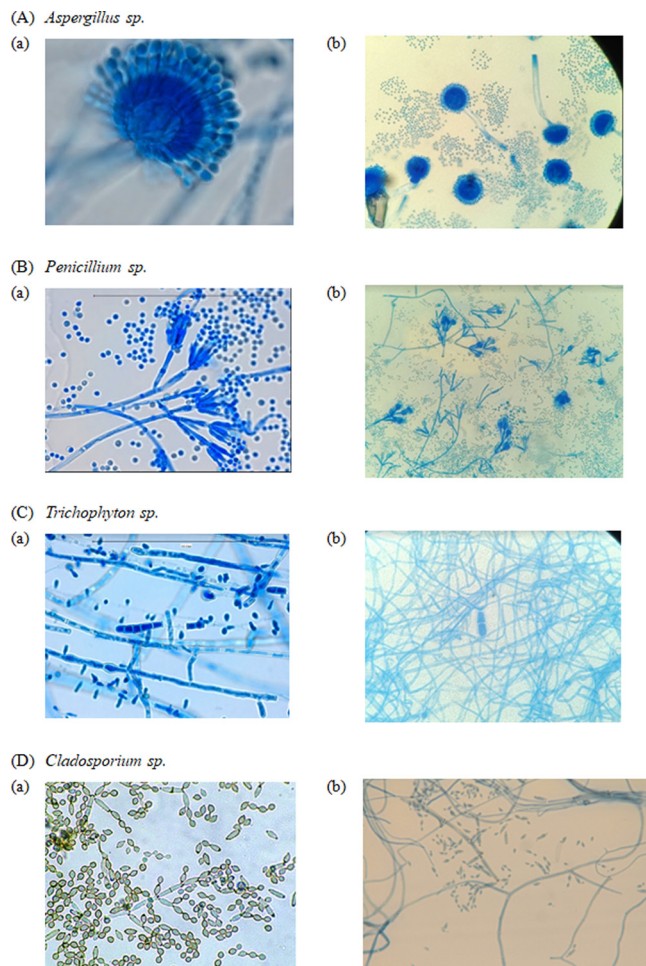

**FIG 3** Representative figures which have been incorrectly identified. Theses figures are obtained from two sources: (a) Google and (b) photomicrographs of touch-tape preparation of fungal culture stained with LPCB and taken using a high-power objective (40×) on a compound light microscope.

This article presents several distinct features. First, we utilized transfer learning to analyze microscopic morphology images obtained from touch-tape technique slides, a common and straightforward diagnostic method in clinical settings. Although slide culture allows for observing more shapes and structures, it requires more time, special conditions, and skilled personnel, making it labor-intensive. Therefore, if microscopic images obtained from Scotch tape preparation can accurately identify filamentous mold, they will be more practical for clinical use.

Second, we incorporated a soft attention mechanism to enhance classification accuracy. Specifically, the test accuracy of our MobileNetV3 Large model in classifying genera

**TABLE 7** Performance metrics of five transfer learning pretrained models on classifying six *Aspergillus* species with five-fold stratified cross-validation[a]

| | Without attention | | | | | With attention | | | | |
|---|---|---|---|---|---|---|---|---|---|---|
| Model | Test accu. | Test loss | Macro F1-score (%) | Prec. (%) | Sens. (%) | Test accu. (%) | Test loss | Macro F1-score (%) | Prec. (%) | Sens. (%) |
| ResNet50 | 61.7 | 1.179 | 53.2 | 59.2 | 54.5 | 81.2 | 0.638 | 79.0 | 80.8 | 78.7 |
| DenseNet201 | 71.1 | 0.937 | 62.8 | 65.8 | 63.1 | 81.2 | 0.592 | 78.8 | 79.8 | 78.9 |
| MobileNetV3 Large | 53.1 | 1.379 | 36.6 | 41.4 | 40.3 | 53.6 | 1.524 | 42.7 | 51.6 | 47.3 |
| EfficientNetV2M | 83.5 | 0.565 | 81.4 | **82.5** | 81.8 | 83.9 | 0.575 | 82.3 | 83.5 | 82.5 |
| Xception | **83.5** | **0.544** | **81.5** | 82.0 | **82.4** | **84.5** | 0.565 | **83.3** | **84.0** | **83.5** |

[a]Bold text indicates the best value among the models. Prec., precision; Sens., sensitivity.

**TABLE 8** Accuracy on classifying six *Aspergillus* species[a]

| | Without attention | | | | | | With attention | | | | | |
|---|---|---|---|---|---|---|---|---|---|---|---|---|
| Model | Afl (%) | Afu (%) | An (%) | At (%) | Av (%) | U (%) | Afl (%) | Afu (%) | An (%) | At (%) | Av (%) | U (%) |
| ResNet50 | 27.8 | 38.6 | 90.1 | 68.6 | 36.3 | 65.9 | 66.2 | 68.1 | 89.1 | 90.7 | 71.9 | 83.2 |
| DenseNet201 | 32.3 | 51.2 | 89.7 | 75.8 | 48.2 | 81.3 | 69.4 | 66.8 | 88.4 | 88.5 | 75.4 | 84.6 |
| MobileNetV3 Large | 22.4 | 17.8 | 81.8 | 56.8 | 13.0 | 68.6 | 27.2 | 19.2 | 87.1 | 55.6 | 35.8 | **58.8** |
| EfficientNetV2M | **76.4** | 71.0 | **94.0** | **93.0** | 73.6 | **82.5** | 73.9 | 78.2 | 90.7 | **93.8** | 76.1 | **82.2** |
| Xception | 72.9 | **73.1** | 93.4 | 92.0 | **81.3** | 81.4 | 73.1 | **78.6** | **92.4** | 92.9 | **82.0** | 81.9 |

[a]Bold text indicates the best value among the models. Afl, *A. flavus*; Afu, *A. fumigatus*; An, *A. niger*; At, *A. terreus*; Av, *A. versicolor*; U, unclassified *Aspergillus* sp.

improved by 21% with the addition of attention. In comparison, the test accuracy of our Xception model in identifying *Aspergillus* species increased by only 1%. Although the percentage increase may appear small for *Aspergillus* species identification, the absolute number of correctly classified images is still noteworthy, especially when dealing with large data sets.

This study highlights the benefits of merging advanced technology with traditional laboratory practices to diagnose filamentous fungi in clinical settings. It is the first instance of utilizing an attention-based deep-learning model to classify images of filamentous fungi. Our approach is unique because we worked with a medical mycology laboratory to develop a model that seamlessly integrates into routine workflows. Furthermore, we were the first to leverage the expertise of medical technologists in image classification of filamentous fungi, opening up possibilities for developing AI-based microscopy image diagnostic tools.

## MATERIALS AND METHODS

**Classification scheme.** The most frequently isolated filamentous fungi in our laboratory were *Aspergillus* spp., *Cladosporium* spp., *Penicillium* spp., and *Trichophyton* spp. Together, they made up around 75% of all isolates. Within the *Aspergillus* spp., we isolated *A. flavus*, *A. fumigates*, *A. niger*, *A. terreus*, and *A. versicolor*, which accounted for 65% of all *Aspergillus* isolates. Therefore, we decided to classify these four genera and the genus *Aspergillus* into five common species. Any species that do not belong to these four genera or the five *Aspergillus* species were classified in the "unclassified" group.

**Fungal microscopic image data set.** To improve the model's performance by increasing the quantity and diversity of images, we merged two data sources: Google images and microscopic images produced in our laboratory.

Initially, we searched for fungal species on Google images and downloaded publicly available images. Next, we took microscopic images of fungal strains prepared in our lab by obtaining slides from the 310 fungal isolates collected from Kaohsiung Veterans General Hospital from 2020 to 2021. The fungal isolates were grown on potato dextrose agar plates and incubated at 25°C. Microscopic images were taken from well-grown colonies using touch-tape preparation and lactophenol cotton blue (LPCB) staining. The images were captured using a DM2500 microscope (Leica Microsystems, Germany) with 40-fold magnification and an EOS 60D camera (Canon, Japan). We recorded 10 to 20 photos per slide with a characteristic morphology, and one slide was taken for each sample. Finally, two experienced medical technologists interpreted all image labels included in this study.

**Convolutional neural network.** The Python code used in Google Colab and Kaggle was utilized to perform fungal image classification using a CNN and transfer learning. First, the CNN was trained with the TensorFlow Core 2.8.0 Python API. The next step was to classify *Aspergillus* at the species level. The entire process was based on transfer learning with an attention mechanism illustrated in Fig. 4A.

**(i) Feature learning.** We employed pretrained models such as ResNet-50 (37), DenseNet-201 (38), MobileNetV3 Large (39), EfficientNetV2 (40), and Xception (41), which were already trained on ImageNet

**TABLE 9** Confusion matrix of *Aspergillus* species classification results of the test set for the Xception model

| | Predicted classification (*n*) | | | | | | | |
|---|---|---|---|---|---|---|---|---|
| *Aspergillus* species | *A. flavus* (*n* = 109) | *A. fumigates* (*n* = 214) | *A. niger* (*n* = 318) | *A. terreus* (*n* = 274) | *A. versicolor* (*n* = 108) | Unclassified (*n* = 465) | Sensitivity (%) | Specificity (%) |
| *A. flavus* (*n* = 111) | 81 | 5 | 9 | 7 | 1 | 8 | 73.0 | 98.0 |
| *A. fumigatus* (*n* = 238) | 5 | 187 | 2 | 24 | 0 | 20 | 78.6 | 97.8 |
| *A. niger* (*n* = 302) | 6 | 2 | 279 | 2 | 0 | 13 | 92.4 | 96.7 |
| *A. terreus* (*n* = 227) | 3 | 3 | 1 | 211 | 1 | 8 | 93.0 | 95.0 |
| *A. versicolor* (*n* = 117) | 6 | 3 | 0 | 0 | 96 | 12 | 82.1 | 99.1 |
| Unclassified (*n* = 493) | 8 | 14 | 27 | 30 | 10 | 404 | 81.9 | 93.9 |

**(A)** Transfer learning framework with attention mechanism

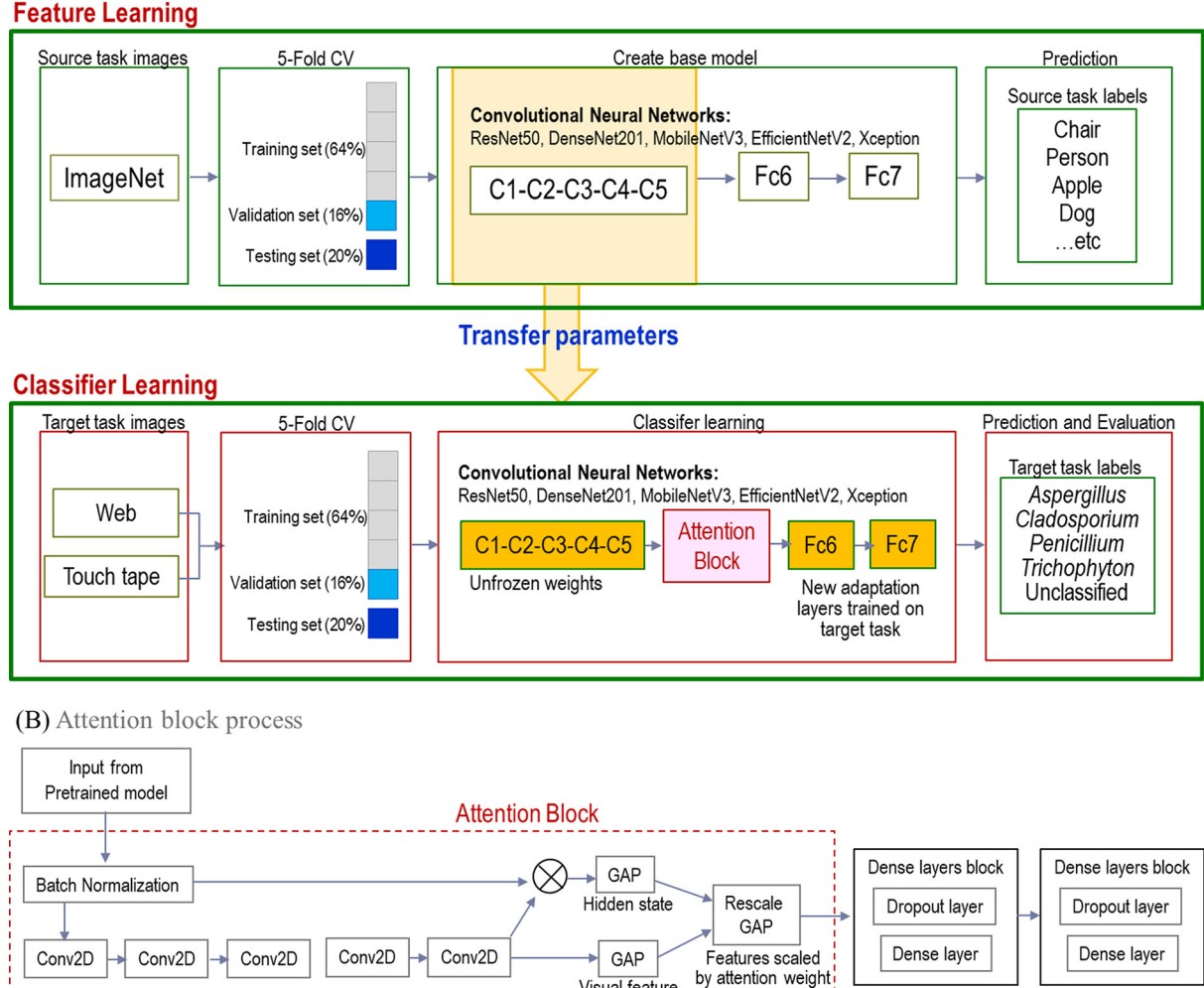

**FIG 4** Panel A depicts the framework that utilizes transfer learning with an attention mechanism. The process begins with the training of CNN models on a large data set, specifically ImageNet. Next, the transfer learning approach is used to transfer knowledge from pretrained weights to the target task and fine-tune the mode. Additionally, the figure displays the introduction of attention mechanism and data adaptation layers, specifically the fully connected layers Fca and Fcb. The process within the attention block is emphasized in panel B, where the attention mechanism and data adaptation layers are employed to improve the model's performance. (A) The framework that utilizes transfer learning with an attention mechanism. (B) The process within the attention block.

and served as TensorFlow ML framework checkpoint files. ImageNet comprises 1,200,000 training and 50,000 test images, categorized into 1,000 object classes.

**(ii) Classifier learning with a soft attention mechanism.** To fit the images into the pretrained ImageNet model, they were resized to 224 × 224 pixels. For this study, a five-fold cross-validation approach was implemented, which involved several steps. To begin with, the available images were split into five folds in a random and stratified manner, with each fold having an approximately equal number of images. The stratification process was based on the five-class genus or six-class *Aspergillus* sp. Next, one fold was reserved for testing in each of the five cross-validation folds, while the remaining four folds were used for training and validation in an 80:20 ratio, respectively. This process was repeated 5 times, with each fold used once for testing and the other four times for training and validation. Finally, the average performance of the training and testing data set were computed as the evaluation index for the models.

In transfer learning, a new classifier in a different network is trained using the features obtained from a large data set. The process begins by training the base network with the ImageNet data set. The initial layers of the base network are then duplicated in the target network, while the remaining layers of the new classifier are randomly initialized. The duplicated layers are frozen, and the new classifier is trained on data. To improve the transfer learning performance, the target network is fine-tuned by unfreezing the top layers of the initial layers while keeping the bottom layers frozen. This approach enables the new classifier and unfrozen top layers to be trained simultaneously, refining the higher-order feature representation of the first few layers for better relevance to specific defect detection tasks. In this work, 30 layers are unfrozen during the target network training process. The resulting network is then utilized for defect detection.

To evaluate the effectiveness of the classification task, different numbers of unfreezing layers were set based on the recommended transfer learning setting provided by TensorFlow (https://www.tensorflow.org/guide/keras/transfer_learning). The best results were achieved when the pretrained model layers were unfrozen to retrain the model on a new data set, given the inherent differences between the images of the ImageNet data set and the development data set. Moreover, data augmentation was performed using Keras to improve the quantity and diversity of training data. For this purpose, the built-in function of TensorFlow 2.9.2, tf.keras.preprocessing.image, ImageDataGenerator was utilized. Keras 2.9.0 and Scikit-learn 1.0.2 were employed during the process. Online augmentations, including random rotations (180°), horizontal and vertical flips (True), width shift (0.1), height shift (0.1), shear (0.1), fill mode (wrap), and zoom (0.1), were applied to each batch of the development data set.

We have improved image classification accuracy by introducing an attention mechanism into CNN. We chose soft attention over hard attention because it is generally more effective in weighing various features and capturing nuanced and contextual information. Soft attention also provides the flexibility of selecting crucial areas of the image that are essential for classification. In contrast, hard attention is restricted to fixed regions and cannot be optimized, potentially missing out on significant regions. Additionally, soft attention can assign different weights to different image regions, while hard attention can only completely cover one region without any weighting. As a result, soft attention is better at capturing detailed image features and allows for different levels of attention to be assigned to different regions of the image, resulting in a more comprehensive contextual understanding. On the other hand, hard attention considers image regions independent and cannot establish any relationships between them. (42, 43) Figure 4B illustrates the soft attention process.

The top portion of the transfer network with the attention mechanism is enriched with batch normalization and four two-dimensional (2D) convolutional (Conv2D) layers, each with 64, 32, 16, and 1 filter, respectively. All Conv2D layers have a kernel size of (1,1). In addition, two GlobalAveragePooling2D layers are included to enhance network performance and reduce the feature map to a single vector of shape (32, 1024) and the attention map to a single scalar value of shape (1, 32), respectively. The feature map is rescaled using a Lambda layer before concatenating it with the attention map. To improve regularization and alleviate overfitting, two dropout layers are used in the network, with the first having a rate of 0.5 and the second having a rate of 0.25. Each dropout layer is followed by a dense layer, with the first dense layer having 128 units and the second dense layer having several units equal to the number of classes in the problem. The model consists of four Conv2D layers, two GlobalAveragePooling2D layers, two dense layers, and one Lambda layer, using approximately 6.6 million neurons. During training, parameter updates are made using the Adam optimizer with a default learning rate of 0.001 and a decay rate of 0 in Keras. The batch size used is 32, and the detailed network architecture can be seen in Fig. 4B, while the network architecture and parameter summary are provided in Table 2.

**Performance metrics.** We employed a five-fold cross-validation approach to evaluate the performance and reported the five-fold average score. The metrics utilized were accuracy, precision, sensitivity, and macro F1-score, with calculations performed per class for multiclass image classification using a one versus all approach. In addition, precision (also known as the positive predictive value, PPV, in laboratory medicine), sensitivity, and the macro F1-score were also computed. Due to imbalanced class distribution, we used the macro-average over the micro-average for these multiclass tasks to avoid misleading results.

The performance of the classification models was evaluated using the following metrics. In binary classification models, the metrics were computed using the following formulas: TP (true positive) for actual positive is predicted positive, TN (true negative) for actual negative is predicted negative, FP (false positive) for actual negative is predicted positive, and FN (false negative) for actual positive is predicted negative. Then the above-described metrics can be evaluated using the following formula:

$$\text{Accuracy} = \frac{(\text{TP} + \text{TN})}{(\text{TP} + \text{TN} + \text{FP} + \text{FN})}$$

$$\text{Precision} = \frac{\text{TP}}{(\text{TP} + \text{FP})}$$

$$\text{Sensitivity} = \frac{\text{TP}}{(\text{TP} + \text{FN})}$$

$$\text{F1 score} = \frac{2 \times \text{Precision} \times \text{Sensitivity}}{(\text{Precision} + \text{Sensitivity})}$$

For classification models with $k$ classes, let Precision$_i$, Sensitivity$_i$, and Macro $F1 - score_i = \frac{2 \times Precision_i \times Sensitivity_i}{Precision_i + Sensitivity_i}$ denote the precision and sensitivity for class $i = 1, 2, \ldots, k$, respectively.

$$\text{Precision} = \frac{1}{k} \sum_{i=1}^{k} \text{Precision}_i$$

$$\text{Sensitivity} = \frac{1}{k}\sum_{i=1}^{k}\text{Sensitivity}_i$$

$$\text{Macro F1} - \text{score} = \frac{1}{k}\sum_{i=1}^{k}\text{Macro F1} - \text{score}_i$$

Training and validation accuracies, as well as categorical cross-entropy test loss, were used to compare models.

The loss function used in our approach is categorical cross-entropy. In iterative training, an epoch refers to one pass of the entire training data set by the machine learning algorithm. The maximum number of epochs used in our approach is 100, and after each epoch, the model's performance is evaluated on the validation data set. To avoid overfitting, we use early stopping, which stops model training if the categorical cross-entropy loss on the validation set does not decrease for five consecutive epochs. The sum of the categorical cross-entropy loss function calculates the loss of an example by computing the following:

$$\text{Loss} = -\sum_{i=1}^{k} y_i \log \hat{y}_i$$

where $\hat{y}_i$ is the $i$th scalar value in the model output (probability), $y_i$ is the corresponding target value (labeled class is 1, otherwise 0), and $k$ is the number of classes in the model output.

**Data availability.** The Python 3.7.13 scripts and mold images used for this project are available on Google Drive (https://drive.google.com/drive/folders/1w-h8bkJw4xtJU3sxSCe9O_3ttk7fBR0j?usp=sharing) and (https://drive.google.com/drive/folders/1S5Y-CVNWEYe5-eVdlMS4OmxpxYjDpENA?usp=sharing).

## ACKNOWLEDGMENTS

This work was financially supported by Kaohsiung Veterans General Hospital (VGHKS110-130 to T.-S.H.) and the Veterans Affairs Council (VAC111-005-12 to T.-S.H.).

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
