## [Reviewer comments · Microbiology Spectrum]

Microbiology Spectrum

Attention-guided transfer learning for identification of filamentous fungi encountered in the clinical laboratory

Tsi-Shu Huang, Kevin Wang, Xiu-Yuan Ye, Chii-Shiang Chen, and Fu-Chuen Chang

Corresponding Author(s): Fu-Chuen Chang, National Sun Yat-sen University

Review Timeline:

Submission Date:	December 20, 2022
Editorial Decision:	February 2, 2023
Revision Received:	March 21, 2023
Editorial Decision:	April 5, 2023
Revision Received:	April 11, 2023
Accepted:	April 12, 2023

Editor: Paschalis Vergidis

Reviewer(s): Disclosure of reviewer identity is with reference to reviewer comments included in decision letter(s). The following individuals involved in review of your submission have agreed to reveal their identity: Yang Zhang (Reviewer #1)

Transaction Report:

DOI: <https://doi.org/10.1128/spectrum.04611-22>

February 2, 2023

Prof. Fu-Chuen Chang
National Sun Yat-sen University
Department of Applied Mathematics
70 Lien-hai Rd.
Kaohsiung 806
Taiwan

Re: Spectrum04611-22 (Attention-guided transfer learning for identification of filamentous fungi encountered in the clinical laboratory)

Dear Prof. Fu-Chuen Chang:

Thank you for submitting your manuscript to Microbiology Spectrum. I reviewed your manuscript and the comments of the previous reviewers. I invited an additional reviewer that has raised several points, as detailed below. Please address these comments.

Please also respond to the following comment of the previous reviewer, as this has not been adequately addressed: Photos used in this work were taken by technologists who presumably were already aware of the diagnostic features of molds and specifically attempted to capture them. In contrast, this method may be best utilized in situations where trained technologists are unavailable. How would the model perform if images were collected in a semi-random fashion by an untrained operator? Could automated microscopy also be used here?

In your revision, please edit your introduction on MALDI-TOF (Lines 64-68). MALDI-TOF is an indispensable tool in Medical Mycology despite some limitations. Overall, the introductory paragraph on diagnostics can be significantly shortened.

Link Not Available

Sincerely,

Paschalis Vergidis

Journals Department
Reviewer comments:

Reviewer #1 (Comments for the Author):

In this manuscript, the authors proposed an attention-guided convolutional neural network for fungal classification. However, this manuscript has several weaknesses as follows:

1. The structure of the model is unclear. What are the settings of the used CNN? How many convolutional layers are used, what is the kernel size of convolution and pooling, what are the settings of fully connected layers, and how many neurons are used? The authors should provide details of their model so other people can replicate their work.
2. In row 162, the authors use the word "x" to denote multiplication. From rows 214 to 217, the used equations forgot to italicize the variables like "k" and "i". Why do the authors define the metric of Accuracy in two different manners, is there any special meaning in this? The authors should reformat the equations to meet the requirements of the journal.
3. The experiment settings need to be more specified. What are the initial settings of learning rate batch size, decay rate, etc? The authors need to provide this information to make their results reproducible.
4. What is the evidence of the "color has no significance in fungal image classification" mentioned in row 159. The authors should provide evidence either from existing literature or by conducting experiments on their own.
5. The reference quality needs to be improved. There are mistakes in the references, for instance, in references 25, 29, 30, etc, the authors forgot to update the reference with their published journal names. Also, some previous related works should be cited. For instance, <https://doi.org/10.1093/bioinformatics/btaa513>.
<https://doi.org/10.1093/gigascience/giab040>.
<https://doi.org/10.1016/j.tim.2021.01.006>.
6. What are the details of using ImageNet to train CNN? The authors said that "We used ImageNet to train CNNs to learn good general-purpose features" in row 154, but no details are mentioned in the following contexts. If the authors simply use the pre-trained model on GitHub, then they should specify it and provide the link, otherwise, details of how they use the ImageNet dataset to train the model are required.
7. The novelty in this manuscript is limited. The use of transfer learning is very common, many studies would use ImageNet pre-trained model weights as a starting point rather than training from scratch. If a different transfer learning approach is used, the authors should outline it to demonstrate their novelty. In addition, the assumed utility of the attention module mentioned in rows 100 to 102 is not validated. The effectiveness of this attention module is not tested, the authors did not perform any ablation study to validate the effectiveness of this module. In addition, the authors did not specify which types of attention are used, is it self-attention or some more traditional attention mechanisms? The results demonstrated in this article are not enough to prove that this idea is novel. More experimental results and analysis of different models are required, such as an ablation study, more specific case studies are needed for each model, what are the advantages and disadvantages of each model, and what might be the possible reasons for their performance differences? The authors should provide their insights rather than simply demonstrate the results.
8. The format of references is superscripted in the beginning and the authors forget to do this in the following paragraphs. In row 98, the position of citation (27) is wrong. In row 114, the citation of (31) is mistakenly inserted into the word "species". In row 125, the word "Trichophyton" is not italicized.
9. In row 161, the author uses NTSC to abbreviate National Television System Committee, but this abbreviation is not used in the following contents. The authors should only use the abbreviations when it is needed in the following contents. The authors should have re-stated the abbreviation of CNN in the text instead of following the abstract because the abstract is usually considered a separate unit.
10. The pictures in this manuscript are of low quality. The provided flowchart has spelling mistakes such as spelling "evaluate" as "evaluae" or "task" as "trask". In addition, the text with red wavy lines indicates the authors took a snapshot over the PowerPoint instead of exporting the image. In figure 2, the font color of the title is mistakenly set to gray and the numbers in the chart add up to 100.1% which is more than 100%. In the Aspergillus category, the font color is mistakenly set to black. And in the subcategories of Aspergillus, there is no indication of what the purple part is.

Staff Comments:

Preparing Revision Guidelines

To submit your modified manuscript, log onto the eJP submission site at <https://spectrum.msubmit.net/cgi-bin/main.plex>. Go to

Author Tasks and click the appropriate manuscript title to begin the revision process. The information that you entered when you first submitted the paper will be displayed. Please update the information as necessary. Here are a few examples of required updates that authors must address:

Please return the manuscript within 60 days; if you cannot complete the modification within this time period, please contact me. If you do not wish to modify the manuscript and prefer to submit it to another journal, please notify me of your decision immediately so that the manuscript may be formally withdrawn from consideration by Microbiology Spectrum.

Response to Editor

Re: Spectrum04611-22

To the Editor

We appreciate the valuable critique you provided for our manuscript. My co-authors and I are grateful for the opportunity to address the reviewer's comments and make the necessary revisions to the manuscript.

Editor

1. As the editor pointed out, this approach can be especially valuable when skilled technologists are unavailable. We included an "other" group in our study to address this issue. This can be beneficial in cases where an inexperienced medical technologist cannot identify which microscopic fields contain morphological characteristics that are useful for classification or in situations where an automated slide scanning and image retrieval system is used in the future. In such a scenario, multiple images could be taken from a single slide. Any pictures that do not exhibit distinctive features or fail to fit into predetermined classes would be assigned to the "other" category. By analyzing all images from the slide as a whole, the sample can be accurately classified if any precisely classified images are present. However, it is crucial to emphasize that further external validation is required to ensure the classification's accuracy. (p. 23, Line 363-368)
2. We have rewritten the introduction to MALDI-TOF, and the opening paragraph on diagnostics has been considerably condensed. (p. 4-5)

Reviewer #1 (Comments for the Author):

1. Please refer to Figure 1 (b) for the detailed network architecture and Table 1 for the network architecture and parameters summary. The model with attention architecture consists of 4 Conv2D layers, with 64, 32, 16, and 1 filters, respectively. All the Conv2D layers have a kernel size of (1,1). The model also includes two GlobalAveragePooling2D layers, which reduce the feature map to a single vector of shape (32, 1024) and the attention map to a single scalar value of shape (32, 1), respectively. The Lambda layer is used to rescale the feature map before concatenating it with the attention map. The first Dense layer has 128 units, and the second Dense layer has a number of units equal to the number of classes in the problem. Therefore, the model has a total of 4 Conv2D layers, 2

GlobalAveragePooling2D layers, 2 Dense layers, and 1 Lambda layer. The total number of neurons in the model is approximately 6.6 million.

2. The equations were reformatted and met the requirements of the journal. The same formula for Accuracy was removed.
3. Please refer to Figure 1 (b) for the detailed network architecture and Table 1 for the network architecture and parameters summary. The batch size is 32. We use the default values for the optimizer "Adam", such as learning rate of 0.001, and decay rate of 0, were used as described in Lines 210-213.
4. We thank the reviewer for the valuable suggestion that led us to retrain our CNN classification model using the RGB color mode. Our experimentation has shown that RGB mode is more effective than grayscale mode. Through this process, we have also understood that grayscale images have only one color channel. In contrast, color images have three color channels, resulting in different classification results when categorizing objects based on pixel values. This difference can impact feature extraction and classification accuracy. Once again, we appreciate the helpful feedback provided by the reviewer.
5. We improved the citation quality of conference papers in Endnote by correctly formatting the citations under the appropriate reference type. In addition, the previously published works were referenced.
6. The procedure for training CNNs using ImageNet was described in the text (p. 11-12, Line 164-185). We made use of five pre-trained models stored in TensorFlow ML framework checkpoint files. (p. 10, Line 147-151)
7. (a) This article presents several distinct features. Firstly, in the "discussion" section (p. 24, Line 376-382) of the manuscript, we explained how we utilized transfer learning to analyze microscopic morphology images obtained from touch-tape technique slides, which is a common and straightforward diagnostic method in clinical settings. Secondly, we incorporated a soft attention mechanism to enhance classification accuracy. These unique features provide a valuable contribution to the application of transfer learning in clinical medicine, despite the use of common techniques. Therefore, we assert that this paper has substantial value and makes a valuable contribution. (b) In this study, we utilized soft attention and compared models with and without attention to evaluate the effectiveness of the attention module. This methodology enables us to show the influence of integrating attention on enhancing model performance. (c) The benefits and drawbacks of each model have been discussed, along with potential factors contributing to performance disparities, which are based on the unique attributes of the images. Furthermore, we have examined the correlation between the strengths and limitations of each model and the inherent characteristics of the

images. (p. 20-21, Line312-322)

8. The references have been updated to the correct format and all mold genus and species names in the text are now in italics.
9. In the revised manuscript, the usage of full names and abbreviations has been double-checked and confirmed.
10. (a) The spelling in the figures has been verified and corrected, and the quality has also been improved. (b) Each number in every category was rounded to one decimal place, which resulted in a total sum of 100.1%. However, the total count for each category is correct.

April 5, 2023

Prof. Fu-Chuen Chang
National Sun Yat-sen University
Department of Applied Mathematics
70 Lien-hai Rd.
Kaohsiung 806
Taiwan

Re: Spectrum04611-22R1 (Attention-guided transfer learning for identification of filamentous fungi encountered in the clinical laboratory)

Dear Prof. Fu-Chuen Chang:

You adequately addressed the comments of the reviewer. You added a group to the study referring to images taken by inexperienced technologists or random images lacking distinctive characteristics. I recommend that you define this group in the Methods section clearly. Please also use a name that will characterize this group (Not "other" group).

Thank you for submitting your manuscript to Microbiology Spectrum. As you will see your paper is very close to acceptance. Please modify the manuscript along the lines I have recommended. As these revisions are quite minor, I expect that you should be able to turn in the revised paper in less than 30 days, if not sooner. If your manuscript was reviewed, you will find the reviewers' comments below.

When submitting the revised version of your paper, please provide (1) point-by-point responses to the issues raised by the reviewers as file type "Response to Reviewers," not in your cover letter, and (2) a PDF file that indicates the changes from the original submission (by highlighting or underlining the changes) as file type "Marked Up Manuscript - For Review Only". Please use this link to submit your revised manuscript. Detailed instructions on submitting your revised paper are below.

Link Not Available

Sincerely,

Paschalis Vergidis

Reviewer comments:

Reviewer #1 (Comments for the Author):

The quality of Figure 1 could be improved, for example by aligning the arrows and adjusting the typography and colour scheme. Some formatting and details need further polishing.

Preparing Revision Guidelines

Please return the manuscript within 60 days; if you cannot complete the modification within this time period, please contact me. If you do not wish to modify the manuscript and prefer to submit it to another journal, please notify me of your decision immediately so that the manuscript may be formally withdrawn from consideration by Microbiology Spectrum.

Response to Editor

Re: Spectrum04611-22

To the Editor

Thank you very much for your valuable feedback. We hope that this version will meet your expectations.

Editor

1. We have renamed the 'others' group as the 'unclassified' group and provided a definition of this change in the 'material and methods' section. We have also made the necessary modifications to Figure 2, tables, and the relevant section in the text. (p. 9, Line 127-128)

Reviewer #1 (Comments for the Author):

1. Fig. 1 has been adjusted in terms of arrow, grid, and overall text layout to make it look more tidy and organized.

April 12, 2023

Prof. Fu-Chuen Chang
National Sun Yat-sen University
Department of Applied Mathematics
70 Lien-hai Rd.
Kaohsiung 806
Taiwan

Re: Spectrum04611-22R2 (Attention-guided transfer learning for identification of filamentous fungi encountered in the clinical laboratory)

Dear Prof. Fu-Chuen Chang:

Your manuscript has been accepted, and I am forwarding it to the ASM Journals Department for publication. You will be notified when your proofs are ready to be viewed.

Sincerely,

Paschalis Vergidis
Editor, Microbiology Spectrum
